# Resource availability, utilisation and cost in the provision of critical care in Tanzania: a protocol for a systematic review

Joseph Kazibwe [ID],[1] Hiral A Shah,[1,2] A Kuwawenaruwa,[3] Carl Otto Schell [ID],[4,5] Karima Khalid,[3,6] Phuong Bich Tran,[7] Srobana Ghosh,[8] Tim Baker [ID],[4] Lorna Guinness [ID] [8,9]

For numbered affiliations see end of article.

**Correspondence to**
Mr Joseph Kazibwe;
j.kazibwe@imperial.ac.uk

## ABSTRACT

**Introduction** Critical care is essential in saving lives of those that are critically ill, however, provision of critical care can be costly and heterogeneous across lower-resource settings. This paper describes the protocol for a systematic review of the literature that aims to identify the reported costs and resources available for the provision of critical care and the forms of critical care provision in Tanzania.

**Methods and analysis** The review will follow the Preferred Reporting Items for Systematic Reviews and Meta-Analyses guidelines. Three databases (MEDLINE, Embase and Global Health) will be searched to identify articles that report the forms of critical care, resources used in the provision of critical care in Tanzania, their availability and the associated costs. The search strategy will be developed from four key concepts; critical care provision, critical illness, resource use, Tanzania. The articles that fulfil the inclusion and exclusion criteria will be assessed for quality using the Reference Case for Estimating the Costs of Global Health Services and Interventions checklist. The extracted data will be summarised using descriptive statistics including frequencies, mean and median of the quantity and costs of resources used in the components of critical care services, depending on the data availability. This study will be carried out between February and November 2021.

**Ethics and dissemination** This study is a review of secondary data and ethical clearance was sought from and granted by the Tanzanian National Institute of Medical Research (reference: NIMR/HQ/R.8a/Vol. IX/3537) and London School of Hygiene and Tropical Medicine (ethics ref: 22866). We will publish the review in a peer-reviewed journal as an open access article in addition to presenting the findings at conferences and public scientific gatherings.

**PROSPERO registration number** The protocol was registered with PROSPERO; registration number: CRD42020221923.

### Strengths and limitations of this study

► This study will follow the Preferred Reporting Items for Systematic Reviews and Meta-Analyses guidelines thereby being the first systematic review of the literature around the costs and resources used in critical care and forms of critical care in a lower resource setting such as Tanzania.

► The search strategy broadly and comprehensively includes studies on critical care and costs of inpatient care in Tanzania increasing chances of including all published studies on the subject.

► This study has a fully developed population, intervention, comparison and outcome framework.

► The study will include only articles published in English.

► There is a chance that there may be limited relevant data on Tanzania in that there may also be a publication bias where published literature focuses on specific diseases (eg, malaria and pneumonia).

countries (LMICs) struggling to meet demand for critical care services.[1 2] This has led to the care of critical illness becoming an urgent issue and a point of focus in global health with stakeholders providing recommendations on what interventions should be scaled up, while knowledge on the potential costs involved remains sparse.[1 3 4]

Critical care entails the care given to a patient in need of specialist monitoring, treatment and attention, for example, for a patient with a life-threatening illness or injury.[5] Over the years, there have been great advances in the provision of critical care, for example, invasive and non-invasive monitoring techniques, mechanical ventilation (MV) and renal replacement therapy, among others which have resulted in reduced mortality rates over time among patients with critical illness especially in high-income countries.[6]

## BACKGROUND

The COVID-19 pandemic is now synonymous with being a critical care crisis with both high-income and low-income and middle-income

Despite the advances made in the provision of critical care in LMICs,[7–9] the limited availability of intensive care units (ICUs) and the low numbers of trained personnel in such settings persist. In addition, the frequent power cuts and inadequate supply of oxygen in some areas make advanced care challenging. The equipment in an ICU can be expensive and may not be affordable or available in many LMICs with an already constrained healthcare budget.[10 11] In a global mapping of ICU bed capacity, the ICU bed density in the majority of sub-Saharan African countries was reported to be below 1.0 ICU bed per 100 000 population as compared with more than 25 in Germany and the USA.[12] Even where ICUs are available in LMICs settings, the services offered are expensive, making them inaccessible and unaffordable for the majority of critically ill patients. This limited availability of critical care services in LMICs has been linked to the high ICU related deaths of between 30% and 80% of all ICU admissions.[8] In fact, most critically ill patients are cared for in general hospital wards.[13 14]

Given the constraints that LMICs face when having to provide critical care, recent research has proposed the concept of essential emergency and critical care (EECC), which focuses on providing low-cost care for critical illness in any place within the hospital.[1 9] EECC consists of effective life-saving actions of low cost and complexity, is appropriate for all hospital settings and for all critically ill patients irrespective of their age, sex, location or underlying medical condition.[9] EECC has been proposed as a key component in universal health coverage. Research is ongoing into the policy and economic aspects of EECC in Tanzania and Kenya to inform EECC implementation in these and other low-resourced health system settings.

To inform this work, it is necessary to review the existing knowledge about the current resource availability, utilisation and cost in the provision of critical care in Tanzania. Several studies have been conducted to estimate the resource use and cost of critical care globally but the majority of these estimates are confined to high-income countries and may not be generalisable to the LMICs settings.[15–17] Even studies that look at the resource use and cost of critical care, whose findings are reported as international estimates, focus on high-income countries, for example, Germany, Hungary, UK and France.[18] In addition, the majority of studies focus on the costs incurred in an ICU only[17] which may not reflect the resources used for all critically illness, especially in LMICs. Here, we describe the protocol of a systematic review that will identify the existing evidence on the quantification of current resources available and the economic costs in the provision of critical care in Tanzania.

## Aim

This paper describes the protocol for a systematic review of the literature that aims to identify the reported costs and resources available for the provision of critical care and the forms of critical care provision in Tanzania.

**Table 1** Population, intervention, comparator, outcome framework used to inform the search strategy

| Element | Description |
|---|---|
| Population | Any patient in need of critical care |
| Intervention | Inpatient services that could form part of critical care provision |
| Comparison | No critical care |
| Outcomes | Critical care services available<br>Cost of the different inputs to critical care provision per patient<br>Quantity and type of resources used in provision of critical care per patient |

## Research questions

1. What are forms of critical care provided in the health system in Tanzania?
2. What resources are reported to be available and have been utilised in the provision of critical care in Tanzania?
3. What are the reported costs of available resources (including oxygen supplies) used in critical care provision in Tanzania?

## METHODS

The systematic review will follow the Preferred Reporting Items for Systematic Reviews and Meta-Analyses (PRISMA) guidelines.[19]

The review will use a population, intervention, comparator, outcome (PICO) framework to enable a comprehensive and unbiased search of existing literature and identify the key search terms[20] (see table 1) . Critical care as a concept does not have an internationally accepted definition. The review defines critical care as the components of care that support a critically ill patient's vital organ functions where critical illness is a state of ill health with vital organ dysfunction, a high risk of imminent death if action is not taken and the potential for reversibility. A critically ill patient is one with acute need for life saving organ support. Critical care is ideally team based and multiprofessional and can be given in any location. Critical care can be high intensity, for example, MV, requiring highly specialised services, or it can be low intensity resuscitative and supportive care, for example, oxygen provided using a face mask.

## Definitions

Forms of critical care are the different levels of critical care services (according to level of advancement) that can be offered to a critically ill patient that can range from basic services like oxygen therapy in general wards to MV in ICUs.

Resources (in this study) are the physical items, material or equipment used in the provision of critical care in a given setting. Resources will be classified using a standard of classification, that is; by input (human resources,

consumables, etc) or by activities (diagnostics, bed days, etc) depending on the data available.

## Search strategy

Electronic databases (Medline (Ovid), Embase (embase.com) and Global Health) will be systematically searched for articles published between 2010 and present. A 10-year time frame will be considered to ensure that findings reflect the present-day resource utilisation paradigm in critical care provision as resources used and their costs vary with the nature of technology and clinical guidelines in place. Bibliographies of included articles will be reviewed to find relevant articles that fulfil the inclusion criteria but not yet identified. Authors whose abstracts will be considered for inclusion, but the full text is not accessible online, will be contacted and requested for article access. Google will also be used to search for published articles that may not have been indexed within the databases. As internet search engines typically return several thousand results, the searches will be restricted to the first fifty hits and links to potentially relevant material will be accessed.[21] The search will be done in anonymous mode to ensure that we do not pick up searcher's embedded preferences. In addition, the involvement of Tanzanian experts in health economics and critical care will be part of the study team to ensure that no obvious articles or reports are missed.

Due to the complex nature of critical care, a combination of strategies was used to identify key search terms that can capture critical illness and critical care. First, the core concepts that we will search on were identified through the PICO framework: critical care (inpatient services that constitute critical care provision), critical illness, resource use (resource utilisation and cost), setting (Tanzania). Subsequently, to identify the search terms, we reviewed the MESH terms related to the key concepts in Medline, consulted within our expert group and scoped our existing critical care and costing libraries. We also include leading causes of death in Tanzania in the search string (eg, neonatal disorders, HIV, malaria and tuberculosis,[22] on the premise that these are conditions that have a high risk of critical illness. Variations of the key words will be included in the search string. The concepts and search terms are listed in table 2. Search strategies were tailored to search the three databases. Search strategies for each database are available in online supplemental appendix 1.

## Eligibility criteria

A study will be considered eligible if it is published in English and reports on forms/types of critical care, services offered under critical care, resources and or costs incurred by the health system or health providers in providing critical care in Tanzania. A detailed inclusion

| Table 2 | Key concepts and search terms identified for the search |
|---|---|
| **Concept** | **Search terms** |
| Inpatient services that can be considered to form part of critical care provision | **Critical care:** Critical care, emergency care, critical emergency care, essential emergency and critical care, intensive care, essential critical care, early goal directed therapy, neonatal, acute care, emergency medicine, trauma, emergency medical services, ICU, perioperative care, maternal emergencies, cardiovascular/inotropic support, renal support **General inpatient care:** General ward care, general inpatient care, general care, hospital care **Oxygen therapy:** Oxygen therapy/provision, artificial ventilation, mechanical ventilation non-invasive ventilation, continuous positive airway pressure, high flow (nasal) oxygen |
| Critical illness | **Critical illness:** Critical illness, sick children, acute paediatrics, emergency obstetric care, polytrauma, severe illness, life-threatening illness, acute illness, emergency illness **Leading causes of mortality in Tanzania:** HIV, malaria, TB, sepsis, trauma, burns, pneumonia, emergency surgery, shock, haemorrhage, respiratory failure, coma, unconsciousness, meningitis, choking, anuria, acute kidney injury |
| Resource use | Cost, expen, spending, invest Financ-, financial burden, financial impact, financial consequence, economic, economic burden, economic impact, economic consequence Direct cost, indirect cost, medical cost, non-medical cost, nonmedical cost, opportunity cost Resource, resource use, resource utilisation, resource utilisation, health service use, health service utilisation, health service utilisation Provider cost, health system cost, hospital cost, system cost, provider cost, hospital cost, societal cost, insurance, reimburs, cost of illness, cost analysis, economic modelling |
| Tanzania | Tanzania |

ICU, intensive care unit; TB, tuberculosis.

exclusion criterion can be found in online supplemental appendix 2.

## Study selection

The PRISMA guidelines[23] will be followed in the selection process. The number of articles retrieved will be listed and uploaded to Rayyan QCRI software, which will be used to identify and remove duplicates.[24] Eligibility of identified studies will be assessed independently by researchers LG, HAS, SG, PBT and JK by first reviewing the title and abstract. All conflicts will be discussed and agreed on through consensus with third researcher (LG or HAS) when applicable. The eligibility of each of the remaining studies will be decided on through full text review of each paper by at least two researchers (SG, JK, HAS, PBT and LG).

## Data extraction process

A data extraction form including author, year of publication, context (location, setting—urban or rural, type of facility, level of facility), critical care services offered (including special critical care services), critical care equipment available, costing perspective, costing year, currency used, type of provider, payer, source of cost data, costing time frame, direct medical costs, resources used, cost ingredients will be developed.

Simultaneously during the data extraction process, studies will be assessed for similarity in cost ingredients to be included in the quantitative (cost) synthesis.

## Risk of bias in individual studies

The quality of the included articles will be assessed using the Reference Case for Estimating the Costs of Global Health Services and Interventions,[25] which provides a framework for quality assessment and data extraction. While standard practice for review of economic evaluations is the Consolidated Health Economic Evaluation Reporting Standards tool, this does not provide sufficient information to assess the quality of cost studies. Any discrepancies will be addressed by a joint re-evaluation of the article among all authors. The appraisal checklist[26] will be used to assess the similarity between the reported costs. For studies that do not report costs, the Newcastle-Ottawa Scale will be used to assess their quality.[27]

## Summary measures and synthesis of results

Studies will be grouped into two: (1) those that report on costs and (2) those that report on forms of critical care. The summary measures used in this study will be descriptive statistics such as frequencies, mean and median critical care services, costs and quantities identified during the data extraction. CIs of the point estimates of costs and resource quantities will be captured when reported by the included studies. Group analysis of the results will be done along level of health facility/hospital, health services offered, cost reported, data sources, study designs, geographical location (rural and urban).

Synthesis of the costs and resources will be performed. Costs, average costs/resources per patient of the different components of critical care will be estimated and converted to 2019 United States dollars (USD) and Tanzanian Shillings (TZS), using the World Bank gross domestic product (GDP) deflators.[28] The costs will be presented per the different components of critical care. A synthesis without meta-analysis checklist[29] will be used to guide the synthesis. A detailed checklist is available as an online supplemental appendix 3. The PRISMA checklist has been included too (online supplemental appendix 4).

## Timeline

The study is expected to be completed by 30 November 2021.

## Patient and public involvement

Patients and the public will not be involved in this research, as this study will focus on reviewing secondary publicly accessible data.

# ETHICS AND DISSEMINATION

This study is a review of secondary data, and ethical clearance was sought from and granted by The Tanzanian National Institute of Medical Research (reference: NIMR/HQ/R.8a/Vol. IX/3537) and London School of Hygiene and Tropical Medicine (ethics ref: 22866).

Following completion of the systematic review, we shall assess whether the research aim and questions have been met and answered, respectively, using the Grading of Recommendations, Assessment, Development and Evaluations frame work.[30]

We aim to publish our findings in a peer-reviewed journal as an open access article in addition to presenting the findings at conferences and public scientific gatherings. These findings will also be a basis and provide special input for a cost effectiveness study of EECC in Tanzania.

**Author affiliations**
[1]Department of Infectious Disease Epidemiology, Imperial College London, London, UK
[2]Department of Infectious Disease Epidemiology, Center for Global Development, London, UK
[3]Health System Impact Evaluation and Policy Unit, Ifakara Health Institute, Ifakara, United Republic of Tanzania
[4]Department of Global Public Health, Karolinska Institutet, Stockholm, Sweden
[5]Department of Global Public Health, Uppsala University, Uppsala, Sweden
[6]Department of Anaesthesia and Critical Care, Muhimbili University of Health and Allied Sciences, Dar es Salaam, United Republic of Tanzania
[7]Department of Family and Population Health, University of Antwerp, Antwerpen, Belgium
[8]Global Health Department, Center for Global Development, London, UK
[9]Global Health & Development, London School of Hygiene and Tropical Medicine, London, UK

**Contributors**  All authors made sufficient contribution to the development of manuscript in line with ICMJE criteria. Below is the detailed breakdown of their contribution. Concept and design: JK, HAS, LG and PBT. Development of first draft: JK. Acquisition of data: N/A. Analysis and interpretation of data: N/A. Drafting of the manuscript: JK, LG, HAS, SG, PBT, COS, KK and AK. Critical revision of paper for important intellectual content: LG, HAS, SG, PBT, TB, COS, KK and AK. Statistical analysis: N/A. Provision of study materials: JK, TB, LG, PBT, COS, KK and AK.

Obtaining funding: TB and COS. Administrative and technical support: JK, LG, HAS, SG, PBT, TB, COS, KK and AK. Supervision: LG and HAS.

**Funding** This work was supported by the Wellcome Trust [221571/Z/20/Z], as part of the 'Innovation in low-and middle-income countries' Flagship.

**Competing interests** None declared.

**Patient consent for publication** Not required.

**Provenance and peer review** Not commissioned; externally peer reviewed.

**ORCID iDs**
Joseph Kazibwe http://orcid.org/0000-0002-8315-1503
Carl Otto Schell http://orcid.org/0000-0002-7904-1336
Tim Baker http://orcid.org/0000-0001-8727-7018
Lorna Guinness http://orcid.org/0000-0002-1013-4200

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
