## [Reviewer comments · BMJ Open]

ARTICLE DETAILS

TITLE (PROVISIONAL)	Resource availability, utilization and cost in the provision of critical care in Tanzania: A protocol for a systematic review
AUTHORS	Kazibwe, Joseph; Shah, Hiral A.; Kuwawenaruwa, A; Schell, Carl Otto; Khalid, Karima; Tran, Phuong Bich; Ghosh, Srobana; Baker, Tim; GUINNESS, LORNA

VERSION 1 – REVIEW

REVIEWER	Vervoort, Dominique Johns Hopkins University Bloomberg School of Public Health
REVIEW RETURNED	26-Apr-2021

GENERAL COMMENTS	The authors present their protocol for a systematic review on the resource utilization, costs, and practices of critical care in Tanzania. I applaud the authors for their important and timely work. I have some comments to improve their manuscript: Major Comments: 1. The “Strengths and Weaknesses of the Proposed Study” section suggests this review will “provide an overview of the current evidence base .. in Tanzania and other low resource settings.” However, the study targets only Tanzania (as any other settings are included in the exclusion criteria), and thus cannot make claims of providing an overview of other variable-resource contexts. This should be adjusted.2. Further, the “Strengths and Weaknesses of the Proposed Study” section suggests there is only one weakness, being that articles are only sought in English. Can the authors not identify any other weakness? (For example, data availability and quality is a potential issue.)3. The authors present their search terms but not the full search strings. It is, therefore, difficult to ensure reproducibility of their search (depending on keywords vs. MeSH, Boolean operators, etc.). Can the authors include their complete and final search strings for the different databases?4. The authors state that “Google and Google Scholar will also be used to search...”: how will this be performed? Google and Google Scholar are difficult to reproduce, thus making the search rather arbitrary without more methodological information. The authors should make sure their methods are reproducible.5. How will cost data be assessed? The authors should account for inflation and differential exchange rates over time, which should be described in the methods.6. Along these lines, the statement “Synthesis of the costs and resources will be performed, where costs can be standardized, ...” is unclear and should be expanded upon.
---

	Minor Comments:  1. Background, third paragraph, global mapping of ICU bed capacity: the cited study has region-, income group-, and country-level data for ICU beds per population. However, the comparison is made between Egypt (an outlier in Africa with many more beds per population than the majority of the continent, including Tanzania) and Monaco (a country with a small population, thus having a density of ICU beds that is far higher than what may be needed). Can the authors instead make the comparison between Tanzania and/or (sub-Saharan) Africa and high-income countries (as a group or individual larger-country examples)? 2. Background, last paragraph, "Even studies that are reported...": This is not a complete sentence and should be rephrased. 3. Search Strategy, second paragraph: "TB" must be introduced before being abbreviated. 4. Data extraction process: the paragraph is not a complete sentence.
--	--

REVIEWER	Gopalan, P. D. University of KwaZulu-Natal, Anaesthesiology & Critical Care
REVIEW RETURNED	07-May-2021

GENERAL COMMENTS	REVIEW Resource availability, utilization and cost in the provision of critical care in Tanzania: A protocol for a systematic review Journal: BMJ Open Manuscript ID bmjopen-2021-050881 Article Type: Protocol The authors are to be complimented for embarking on this study to fill a crucial gap in critical care data in Tanzania. African critical care data in general is sparse and any effort to address this should be welcomed. Overall, this is a well-constructed systematic review which due diligence to all important areas. My primary concern is whether their well-designed systematic review will adequately and appropriately answer their aim. As a systematic review, the expectation is that these data have already been published somewhere. Cost and resource data, especially in the African context have traditionally been poorly documented. Their aim "to identify the reported costs and resources available for the provision of critical care and the forms of critical care provision in Tanzania" may have been better answered by an up-to-date national audit of critical care services. There are a few other issues that may be considered:  • A key research question is "What are forms of critical care provided in the health system in Tanzania?" It is not clear what is meant by "forms of critical care". Does this refer to levels of care e.g. high care units versus intensive care units? Perhaps, some clarity on the definition of "forms of critical care" may be useful in assisting in data extraction and analysis.
---

	 • An additional research question attempts to address “resources”. Again, it is unclear on what resources the review will be focussed. The data extraction form includes fields for critical care equipment available and resources used. A clear definition of resources including categorization into human resources, equipment, pharmaceutical, etc may be useful. • The authors acknowledge the lack of a universal definition of critical care and have included a reasonable overall definition for their review. Considering critical care in terms of the ‘acute need for life-saving organ support’ may make it easier when faced with distinguishing between various patients and between various levels of care. • The authors commendably use the PICO system. This poses some challenges for their review. The population being considered is “any patient in need of critical care”. This broad group is often poorly described in studies. Additionally, a comparison is listed as “no critical care”. It is unclear whether the authors will include this group as it seems that this is not a comparative review. • As part of their search strategy, the authors include oxygen therapy and respiratory support, which I expect is the commonest form of organ support in their setting. It may be prudent for completeness to consider including other organ support e.g. cardiovascular/inotropic support, renal support etc. • The authors correctly place a time restriction for their findings to reflect present day resources. Their chosen period of 10 years must apply due consideration that much could have changed in terms of resources even in that period. • Whilst it is expected that there may not be a great degree of distinction between various specialist critical care services (e.g. cardiac critical care, neurocritical care etc.) in their setting, the data extraction form may benefit from distinguishing between the various specialist critical care services, even if it just for adults versus paediatrics, for example. • Dates and timelines are not noted for the proposed review
--	--

VERSION 1 – AUTHOR RESPONSE

#	Reviewer #1 s’ comments	Authors’ response
1	The “Strengths and Weaknesses of the Proposed Study” section suggests this review will “provide an overview of the current evidence base .. in Tanzania and other low resource settings.” However, the study targets only Tanzania (as any other settings are included in the exclusion criteria), and thus cannot make claims of providing an overview of other variable-resource contexts. This should be adjusted.	Thank you for the comment. We agree that one country review cannot give an overview of the whole income group. We have replaced “overview” with “insight” and stated as follows; “This study will provide an overview on the current evidence base on resources and costs for decision-making in critical care in Tanzania and insight for other low resource settings.” Page 2, Lines 55-57

2	Further, the “Strengths and Weaknesses of the Proposed Study” section suggests there is only one weakness, being that articles are only sought in English. Can the authors not identify any other weakness? (For example, data availability and quality is a potential issue.)	About availability of data, we did a preliminary search and found a number of relevant articles. However, we have agreed to include it as one of the likely challenges. We have included the following; “There is a chance that there may be limited relevant data on Tanzania.....” Page 2, line 63
3	The authors present their search terms but not the full search strings. It is, therefore, difficult to ensure reproducibility of their search (depending on keywords vs. MeSH, Boolean operators, etc.). Can the authors include their complete and final search strings for the different databases?	Search strategies have been included in the appendix. See appendix 1.
4	The authors state that “Google and Google Scholar will also be used to search...”: how will this be performed? Google and Google Scholar are difficult to reproduce, thus making the search rather arbitrary without more methodological information. The authors should make sure their methods are reproducible.	We have removed Google scholar from the search strategy and left only google since majority of articles that are listed by google scholars are in most cases listed in search databases. The search strategy for google will be limited to the first 50 hits as stated in the manuscript; “Google will also be used to search for published articles that may not have been indexed within the databases. As internet search engines typically return several thousand results, the searches will be restricted to the first fifty hits and links to potentially relevant material will be accessed. The search will be done in anonymous mode to ensure that we do not pick up searcher's embedded preferences.”. Page 5, lines 176-177
5	How will cost data be assessed? The authors should account for inflation and differential exchange rates over time, which should be described in the methods.	We have included this information; “Costs, average costs/resources per patient of the different components of critical care will be estimated and converted to 2019 USD and TZS, using the World Bank GDP deflators ²⁸ ” Page 7, lines 235-237
6	Along these lines, the statement “Synthesis of the costs and resources will be performed, where costs can be standardized, ...” is unclear and should be expanded upon.	This has been included here; “Synthesis of the costs and resources will be performed. Costs, average costs/resources per patient of the different components of critical care will be estimated and converted to 2019 USD and TZS, using the World Bank GDP deflators ²⁸ . The costs will be presented per the different components of critical care.” Page 7, lines 235-237

	Minor comments	
1	Background, third paragraph, global mapping of ICU bed capacity: the cited study has region-, income group-, and country-level data for ICU beds per population. However, the comparison is made between Egypt (an outlier in Africa with many more beds per population than the majority of the continent, including Tanzania) and Monaco (a country with a small population, thus having a density of ICU beds that is far higher than what may be needed). Can the authors instead make the comparison between Tanzania and/or (sub-Saharan) Africa and high-income countries (as a group or individual larger-country examples)?	Agreed. We have addressed this by including the following, "In a global mapping of ICU bed capacity, the ICU bed density in the majority of Sub-Saharan African countries was reported to be below 1.0 ICU bed per 100,000 population as compared with more than 25 in Germany and the USA ¹² " Page 3, Lines 105-107
2	Background, last paragraph, "Even studies that are reported...": This is not a complete sentence and should be rephrased.	The sentence has been rephrased and completed as follows; "Even studies that look at the resource use and cost of critical care, whose findings are reported as international estimates, focus on high income countries for example Germany, Hungary, UK and France ¹⁸ " Page 4, Lines 128-129
3	Search Strategy, second paragraph: "TB" must be introduced before being abbreviated	TB has been written in full at the point of first use Page 5, Line 189
4	Data extraction process: the paragraph is not a complete sentence.	The sentence has been completed as follows ".....critical care services offered, critical care equipment available, costing perspective, costing year, currency used, type of provider, payer, source of cost data, costing time frame, direct medical costs, resources used, cost ingredients will be developed." Page 7, line 216
#	Reviewer #2 s' comments	Authors' response
1	My primary concern is whether their well-designed systematic review will adequately and appropriately answer their aim. As a systematic review, the expectation is that these data have already been published somewhere. Cost and resource data, especially in the African context have traditionally been poorly documented. Their aim "to identify the reported costs and resources available for the provision of critical care and the forms of critical care provision in Tanzania" may have been better answered by an up-to-date national audit of critical care services.	Thank you for highlighting this. We agree that cost and resource data in the African context may be scarce and scattered. This is one of the reasons why we are carrying out the review to aggregate all the relevant cost and resources data into one place. We have done some preliminary searches and found some relevant articles on Tanzania. That said, we have mentioned in the weaknesses/limitations of the study that "There is a chance that there may be limited relevant data on Tanzania." This review will provide a good evidence base on whether there is a justified need for a national audit of critical care services.

		Page 2, Line 63
2	A key research question is “What are forms of critical care provided in the health system in Tanzania?” It is not clear what is meant by “forms of critical care”. Does this refer to levels of care e.g. high care units versus intensive care units? Perhaps, some clarity on the definition of “forms of critical care” may be useful in assisting in data extraction and analysis.	Agreed. We have included a definition of the forms of critical care as follows; “Forms of critical care are the different levels of critical care services (according to level of advancement) that can be offered to a critically ill patient that can range from basic services like oxygen therapy in general wards to mechanical ventilation in ICUs” Page 3, Lines 80-82
3	An additional research question attempts to address “resources”. Again, it is unclear on what resources the review will be focussed. The data extraction form includes fields for critical care equipment available and resources used. A clear definition of resources including categorization into human resources, equipment, pharmaceutical, etc may be useful.	Agreed. We have included a definition of resources as follows; “Resources (in this study) are the physical items, material or equipment used in the provision of critical care in a given setting. Resources will be classified using a standard of classification, that is; by input (human resources, consumables etc) or by activities (diagnostics, bed days etc) depending on the data available.” Page 3, Lines 83-86
4	The authors acknowledge the lack of a universal definition of critical care and have included a reasonable overall definition for their review. Considering critical care in terms of the ‘acute need for life-saving organ support’ may make it easier when faced with distinguishing between various patients and between various levels of care.	Thank you for the suggestion. The definition of critical care adopted for this review is intended to try to be as clear and specific as possible to cover all services that comprise critical care. The suggested definition is precise and fits well as a definition for a critically ill patient. We have included it as follows; “A critically ill patient is one with acute need for life saving organ support.” Page 4, Lines 155-156
5	The authors commendably use the PICO system. This poses some challenges for their review. The population being considered is “any patient in need of critical care”. This broad group is often poorly described in studies. Additionally, a comparison is listed as “no critical care”. It is unclear whether the authors will include this group as it seems that this is not a comparative review.	We agree that the patient group may be poorly described in some studies and as such we shall use the definition of critical care to identify the patients that are in need of critical care in case, they have not been well described in a given study. In addition, such patients will also be identified based on the services they are reported to have received. We are not reporting on the comparator but rather this provides the baseline from which costs are measured i.e. what is the incremental cost of providing the critical care for the critically ill patient.
6	As part of their search strategy, the authors include oxygen therapy and respiratory support, which I expect is the commonest form of organ support in their setting. It may be prudent for completeness to consider including other organ support e.g.	Agreed. This has been added as follows; “... cardiovascular/inotropic support, renal support” and adjusted the search strategy in appendix 2

	cardiovascular/inotropic support, renal support etc.	Page 5 and 6, Table 2
7	The authors correctly place a time restriction for their findings to reflect present day resources. Their chosen period of 10 years must apply due consideration that much could have changed in terms of resources even in that period.	Agreed. Thank you.
8	Whilst it is expected that there may not be a great degree of distinction between various specialist critical care services (e.g. cardiac critical care, neurocritical care etc.) in their setting, the data extraction form may benefit from distinguishing between the various specialist critical care services, even if it just for adults versus paediatrics, for example.	Agreed. This is noted and will be captured under critical care services in the data extraction form. This can be elaborated as follows; “....context (location, setting- urban or rural, type of facility, level of facility), critical care services offered (including special critical care services), critical care equipment available, costing perspective,.....” Page 7, Line 214
9	Dates and timelines are not noted for the proposed review	We have included the anticipated timelines as follows; “The study is expected to be completed by 31 st October 2021.” Page 7, Line 241

VERSION 2 – REVIEW

REVIEWER	Gopalan, P. D. University of KwaZulu-Natal, Anaesthesiology & Critical Care
REVIEW RETURNED	03-Jul-2021

GENERAL COMMENTS	Thank you for addressing all the queries raised.
--